

# Prevalence and associated factors of physical-psychological-cognitive multimorbidity in Chinese community-dwelling older adults: a cross-sectional study

Lin Lin[1], Di-fei Duan[2], Linjia Yan[3] and Hai yan He[1]

[1] School of Nursing, Sichuan College of Traditional Chinese Medicine, Mianyang, China
[2] Department of Nephrology, Kidney Research Institute, West China Hospital of Sichuan University, Chengdu, China
[3] The Nethersole School of Nursing Faculty of Medicine, The Chinese University of Hong Kong, Hong Kong, China

Corresponding author
Hai yan He, hhy158846@163.com

## ABSTRACT

**Background:** The rising prevalence of physical-psychological-cognitive (PPC) multimorbidity among older adults poses significant challenges. Understanding its prevalence and associated risk factors is crucial for the development of targeted and effective care strategies.

**Methods:** This cross-sectional study utilized convenience sampling to survey older adults residing in two cities in Sichuan Province and Chongqing, Southwest China, between September 2024 and December 2024. Data were collected using the General Information Questionnaire, EQ-5D-5L, HALFT scale, Patient Health Questionnaire-9, and the 8-item Ascertain Dementia tool. Univariate and multivariate logistic regression analyses were performed to identify predictors of PPC multimorbidity.

**Results:** A total of 437 participants were included, with 75 having PPC multimorbidity and 362 without, resulting in a prevalence of PPC of 17.2%. Social frailty was observed in 17.8%. Multivariate logistic regression identified long-term medication use (OR = 3.24, 95% CI [1.28–8.21]), higher multimorbidity burden (OR = 7.31, 95% CI [3.27–16.36]), social frailty (OR = 3.49, 95% CI [174–7.01]), and lower EQ-5D scores (OR = 0.07, 95% CI [0.02–0.26]) as significant predictors of PPC status (all $p < 0.05$).

**Conclusion:** This study highlights the burden of PPC multimorbidity in older adults in China, with key factors including long-term medication use, advanced multimorbidity, social frailty, and poor quality of life. It calls for a patient-centered care approach that addresses these issues, with future research focusing on larger, diverse samples to guide targeted interventions.

## INTRODUCTION

The escalating global prevalence of multimorbidity, defined as the coexistence of two or more chronic diseases within an individual, presents a formidable challenge to healthcare systems worldwide and substantially compromises the well-being of affected populations (*Ho et al., 2021*). Multimorbidity is highly prevalent among older individuals, especially those over the age of 85, where the frequency can reach up to 82% (*The PLOS Medicine Editors, 2023*). According to projections from the World Health Organization, the proportion of people aged 60 and above is expected to nearly double from 12% to 22% between 2015 and 2050 (*Dogra et al., 2022*). This phenomenon is particularly pronounced among community-dwelling older adults in China, where 70.6% of individuals aged ≥ 65 years experience multimorbidity, significantly exceeding rates observed in younger age groups (*Zhang et al., 2022*). The complex interplay of multiple chronic conditions leads to a cascade of negative consequences, including reduced quality of life (QOL), increased healthcare utilization, heightened disability rates, and ultimately, increased mortality (*Zhao et al., 2021*; *Makovski et al., 2019*).

The psychological burden of chronic age-related diseases significantly influences health outcomes among elderly individuals living in communities across China. Depressive disorders, in particular, show marked urban-rural differences, with higher prevalence rates observed in rural areas compared to urban ones among multimorbid older populations (*Han et al., 2022*). These disparities are driven by a combination of factors, including socioeconomic disadvantages, fragmented healthcare access, and cultural stigmas surrounding mental health (*Yin et al., 2020*). Depression, in this context, not only leads to noncompliance with treatment but also increases the need for acute care, thereby creating a feedback loop that accelerates the progression of chronic diseases (*Li et al., 2022*; *Chen et al., 2023*).

Cognitive dysfunction is another critical aspect of multimorbidity in the aging population of China (*Zhang et al., 2024*). Research indicates that cognitive decline in older adults progresses along a continuum, from mild cognitive impairment to more severe forms such as dementia. The causes of cognitive dysfunction go beyond traditional neurobiological explanations, involving factors shaped by cultural and societal influences (*Ruan et al., 2025*). Moreover, the interplay of systemic inflammation, vascular health, and contextual barriers—particularly limited health literacy and unequal access to cognitive enrichment—creates compounded vulnerabilities, further hindering the ability of elderly individuals to manage their health and engage in treatment (*Wu et al., 2024*).

Social frailty, characterized by decreased social participation and isolation, is increasingly recognized as a significant factor influencing health outcomes in older adults (*Hays et al., 2017*; *Wang et al., 2023*). Social isolation may exacerbate existing health problems and increase vulnerability to new conditions (*Nicholson et al., 2020*). Furthermore, the use of multiple medications, polypharmacy, is very common among individuals with multiple chronic diseases. Polypharmacy increases the risk of adverse drug events, drug interactions, and medication-related problems, further compounding the complexity of managing multimorbidity (*Hays et al., 2017*).

The significant impact of multimorbidity on QOL and health outcomes in older adults highlights the urgent need for comprehensive research to identify the key factors that influence the development and progression of this complex condition (*Ho et al., 2021*). Recent studies on multimorbidity in aging populations have increasingly focused on the co-occurrence of physical, psychological (*e.g.*, depression), and cognitive disorders (*e.g.*, dementia), as well as combinations of these conditions (*Ni et al., 2023*). This "physical-psychological-cognitive" (PPC) multimorbidity is now recognized as a crucial determinant of health outcomes in older adults, with the interaction between these domains playing a significant role in both disease severity and overall well-being (*Du, Liu & Liu, 2024*). Understanding the relationships between these conditions and the factors that influence their progression is essential for the development of targeted interventions (*Zhou et al., 2024*).

This study explores the limited evidence on the prevalence and associated factors of PPC multimorbidity among community-dwelling older adults in China. To address this gap, we posed the research question: What is the prevalence of PPC multimorbidity, and what factors are associated with its occurrence among community-dwelling older adults in China? The findings aim to inform the development of targeted interventions to manage this complex condition more effectively.

## MATERIALS AND METHODS

We used STROBE Checklist for more rigorous study design and improved article quality.

This is a cross-sectional study, in which a total of 437 elderly participants were recruited between September 2024 and December 2024 using a convenience sampling method. The participants were recruited from two communities in Luzhou city, Sichuan Province, and Chongqing city, China. Inclusion criteria: Participants aged 60 or older, with no severe hearing, speech, or cognitive impairments affecting communication, and who voluntarily agreed to participate. Exclusion criteria: Individuals with severe acute or chronic conditions (*e.g.*, terminal illness, advanced cancer) that would hinder study participation, or those unable to complete study procedures independently or with assistance from a research assistant (RA). To ensure the quality and reliability of the data, several quality control measures were implemented throughout the study. First, the research team conducted comprehensive training sessions for three RAs to standardize data collection procedures and minimize observer bias. Furthermore, a structured protocol was followed to ensure consistency in participant recruitment. The recruitment process is summarized in Fig. S1, which details the inclusion/exclusion criteria, screening steps, and final sample composition. This flowchart provides a visual representation of how participants were selected and ensures reproducibility of the sampling strategy.

Rigorous quality control measures ensured complete data (0% missing). Real-time validation during face-to-face interviews, conducted by trained RAs, guaranteed full questionnaire completion and immediate resolution of ambiguities. Data were independently entered into two databases and cross-checked for discrepancies, while regular audits verified protocol adherence. These procedures enhanced data accuracy and maintained the study's scientific rigor.

All procedures involving human participants in this study adhered to the ethical guidelines set by the institutional and national research committees, as well as the principles outlined in the 1964 Declaration of Helsinki and its subsequent amendments, or other comparable ethical standards. Informed written consent was obtained from all participants, and the study received approval from Sichuan University West China Hospital Biomedical Ethics Review Committee (Approval No. 2024(2475)).

## Sample size calculations

The sample size calculation was based on the reported prevalence of PPC multimorbidity, which is 18% in middle-aged and older adults (*He et al., 2024*). Assuming a power of 0.80, a significance level of 0.05, and a margin of error of 5%, the required sample size was determined using PASS 15.0 software. The calculation indicated that a minimum of 430 participants would be necessary for the study.

## Demographic data and covariates

The first section of the questionnaire gathers demographic data, including age, gender, economic status, employment status, educational level, and the presence of chronic diseases. It also collects information on participants' history of alcohol consumption, smoking habits, and physical activity patterns. Additionally, this section assesses the history of long-term medication use, defined as the use of medication for more than 3 months (*Karmali et al., 2020*), including details on the number and frequency of medications.

## The operational definition of PPC multimorbidity

The operational definition of PPC multimorbidity in this study is as follows:

Physical condition: Having at least one physician-diagnosed chronic disease, as self-reported by the participant and verified through medical records or current medication use. Psychological condition: A score of 5 or higher on the Patient Health Questionnaire-9 (PHQ-9), indicating at least mild depressive symptoms. Cognitive condition: A score of 2 or higher on the Ascertain Dementia 8-item (AD-8) questionnaire, suggesting potential cognitive impairment. Participants meeting all three criteria simultaneously were classified as having PPC multimorbidity.

## The HALFT scale

The scale is composed of one letter from each of the words: Help, Participation, Loneliness, Financial, and Talk. This scale is a simple self-report tool for assessing social frailty, consisting of five items with a scoring range from 0 to 5. A score of 0 indicates no social frailty; scores of 1–2 indicate early-stage social frailty; and scores ≥ 3 indicate social frailty. The HALFT scale, utilized to assess social frailty in this study, demonstrated a Cronbach's α coefficient of 0.602. While this value is modest, it is considered acceptable for brief, formative scales comprising a limited number of items. The HALFT scale's brevity and prior validation in Chinese community-dwelling older adult populations support its feasibility and applicability, particularly in settings where minimizing respondent burden and interview duration is essential (*Ma, Sun & Tang, 2018*).

### EQ-5D-5L questionnaire

The Chinese version of the EQ-5D-5L scale consists of five items. Each item is divided into five levels: no problems, slight problems, moderate problems, severe problems, and extreme problems (*Yang et al., 2018*). The first two items assess mobility and self-care, reflecting physical health; the next item, usual activities, reflects social functioning; and the last two items, pain/discomfort and anxiety/depression, reflect mental health. Responses were converted into utility scores using a time trade-off value set, mapping health states onto a scale ranging from −0.158 (worst health) to 1.000 (perfect health). Details are provided in Table S1 (*Luo et al., 2017*).

### Patient health questionnaire-9

Depressive symptoms were assessed using the patient health questionnaire-9 (PHQ-9), which evaluates whether older adults have experienced depressive symptoms over the past 2 weeks. The PHQ-9 includes nine items, each with a four-point response scale ranging from "0" to "3," corresponding to "None," "Few days," "More than half of the days," and "Almost every day," respectively. The total score ranges from 0 to 27, with higher scores indicating more severe depressive symptoms. Depressive symptoms were defined as a PHQ-9 total score ≥ 5, a threshold that corresponds to 'mild depression' in the original validation study by *Kroenke, Spitzer & Williams (2001)*, and is widely used in community-based surveys, including Chinese cohorts of older adults (*Peng et al., 2024*). In this study, the Cronbach's α was 0.869.

### The ascertain dementia-8 questionnaire

The Chinese version of the ascertain dementia-8 (AD-8) questionnaire was used to assess participants' cognitive function, particularly focusing on memory evaluation (*Chen et al., 2018*). The AD-8 is a brief, informant-based tool consisting of eight questions. This scale employs a simple question-and-answer format to quickly determine if a respondent has memory impairments that affect their ability to carry out daily activities. Due to its simplicity and brevity, the AD-8 is well-suited for use by non-specialized primary healthcare providers. The AD-8 has demonstrated strong diagnostic accuracy in distinguishing cognitive impairment from normal cognition (*Galvin et al., 2006*). Scores range from 0 to 8, with scores of 2 or higher indicating memory impairment. The scale is easy to administer and has been shown to have validity and accuracy comparable to the Mini-Mental State Examination (*Mao et al., 2018*). In this study, the Cronbach's α was 0.869.

### Statistical analysis

Descriptive statistics, including median and interquartile range (IQR), were calculated for continuous variables, while frequencies and percentages were calculated for categorical variables. Univariate analysis was conducted using chi-square tests, Fisher's exact tests, and Mann-Whitney U tests. Variables with $p < 0.1$ in the univariate analysis were included in the multivariable logistic regression model, with PPC multimorbidity as the dependent variable. Model selection was based on theoretical considerations and statistical criteria.
IBM SPSS Version 25.0 (IBM Corp., Armonk, NY, USA) was used for statistical analysis, and $p < 0.05$ was considered statistically significant.

## RESULTS

### Demographic characteristics, covariate data, and univariate analysis results

A total of 437 participants were included in this study, comprising 75 in the PPC group and 362 in the non-PPC group. Among the overall sample, 225 (51.5%) were male, and 212 (48.5%) were female. The median age of all participants was 72.0 years (IQR: 65.0–80.0). Social frailty was identified in 78 individuals (17.8%). Health-related quality of life (EQ-5D-5L) had a median score of 0.94 (IQR: 0.73–1.00) across the total sample. The univariate analysis found significant associations between PPC multimorbidity and several factors, including living alone, number of children, weekly moderate-intensity physical activity, long-term medication use, number of medication types, frequency of medication use, number of comorbidities, social frailty, and the EQ-5D-5L trade-off value ($p < 0.05$). Detailed results are presented in Table 1.

### Prevalence of physical, psychological, and cognitive comorbidities

The highest prevalence was observed for chronic diseases (57.9%), followed by cognitive dysfunction (31.6%) and depression (24.0%). The prevalence of comorbidities decreased in the following order: physical-cognitive multimorbidity (29.1%), physical-psychological multimorbidity (21.7%), psychological-cognitive multimorbidity (18.5%), and physical-psychological-cognitive multimorbidity (17.2%). These findings suggest that chronic diseases and their associated comorbidities are widespread among the study population, with a particularly prominent co-occurrence of physical diseases and cognitive dysfunction (Fig. 1).

### Results of multivariate logistic regression analysis

Multivariate logistic regression (Table 2) identified long-term medication use (OR = 3.24, 95% CI [1.28–8.21], $p = 0.013$), higher multimorbidity burden (≥3 conditions: OR = 7.31, 95% CI [3.27–16.36], $p < 0.001$), social frailty (OR = 3.49, 95% CI [1.74–7.01], $p < 0.001$), and lower EQ-5D scores (OR = 0.07, 95% CI [0.02–0.26], $p < 0.001$) as independent predictors of PPC status. The Hosmer–Lemeshow goodness-of-fit test indicated an adequate model fit ($\chi^2 = 10.974$, df = 6, $p = 0.089$) (*Hosmer, Lemeshow & Sturdivant, 2013*). The logistic regression model explained 39.8% of the variance in the dependent variable (Nagelkerke $R^2 = 0.398$).

## DISCUSSION

This is the first study to highlight the significant burden of PPC multimorbidity among community-dwelling older adults in China, revealing varying prevalence rates across different multimorbidity types: physical-cognitive (29.1%), physical-psychological (21.7%), psychological-cognitive (18.5%), and combined PPC multimorbidity (17.2%). Additionally, multivariable analysis identified long-term medication use, advanced

**Table 1 Demographic characteristics, covariate data, and univariate analysis result ($n = 437$).**

|  | PPC group $n = 75$ | Non-PPC group $n = 362$ | Test statistic | p value |
|---|---|---|---|---|
| **Gender** |  |  | 0.397 | 0.529 |
| Male | 44 (58.7) | 181 (50.0) |  |  |
| Female | 39 (41.3) | 187 (50.0) |  |  |
| **Age** | 72.0 (65.0, 82.0) | 72.0 (66.0, 80.0) | 0.220 | 0.826 |
| **Educational years** |  |  | 0.025 | 0.876 |
| ≤6 years | 26 (34.7) | 137 (37.8) |  |  |
| >6–9 years | 29 (38.7) | 118 (32.6) |  |  |
| >9–12 years | 10 (13.3) | 61 (16.9) |  |  |
| >12 years | 10 (13.3) | 46 (12.7) |  |  |
| **Marital status** |  |  | 0.789 | 0.390 |
| Married | 52 (69.3) | 269 (74.3) |  |  |
| Unmarried | 23 (30.7) | 93 (25.7) |  |  |
| **Source of income** |  |  | 3.160 | 0.206 |
| Pension | 51 (68.0) | 221 (61.0) |  |  |
| Children | 14 (18.7) | 103 (28.5) |  |  |
| Others | 10 (13.3) | 38 (10.5) |  |  |
| **Health insurance** |  |  | 1.461 | 0.227 |
| Yes | 68 (90.7) | 340 (93.9) |  |  |
| No | 7 (9.3) | 22 (6.1) |  |  |
| **Individual income monthly** |  |  | 1.240 | 0.265 |
| <2,000 yuan | 18 (24.0) | 52 (14.4) |  |  |
| 2,000–4,000 yuan | 20 (26.7) | 119 (32.9) |  |  |
| >4,000 yuan | 10 (13.3) | 64 (17.7) |  |  |
| Not reported | 27 (36.0) | 127 (35.1) |  |  |
| **Employment status** |  |  |  | 0.549 |
| Employed | 2 (2.7) | 18 (5.0) |  |  |
| Unemployed | 73 (97.3) | 344 (95.0) |  |  |
| **Number of children** |  |  | 13.066* | 0.008 |
| 0 | 2 (2.7) | 12 (3.3) |  |  |
| 1 | 15 (20.0) | 143 (39.5) |  |  |
| 2 | 30 (40.0) | 110 (30.4) |  |  |
| ≥3 | 28 (37.3) | 97 (26.8) |  |  |
| **Living alone** |  |  | 5.109 | 0.024 |
| Yes | 27 (34.9) | 85 (23.5) |  |  |
| No | 48 (65.1) | 277 (76.5) |  |  |
| **Smoking history in the past month** |  |  | 2.366 | 0.306 |
| Yes | 12 (16.0) | 53 (14.7) |  |  |
| No | 13 (17.3) | 93 (25.7) |  |  |
| Never | 50 (66.7) | 216 (59.7) |  |  |

(Continued)

|  | PPC group n = 75 | Non-PPC group n = 362 | Test statistic | p value |
|---|---|---|---|---|
| **Alcohol consumption in the past month** |  |  |  | 0.607* |
| Never | 64 (85.3) | 292 (80.7) |  |  |
| No | 8 (8.0) | 43 (10.2) |  |  |
| Yes | 3 (6.7) | 27 (9.1) |  |  |
| **Moderate-intensity physical activity weekly** |  |  | 13.678 | 0.003 |
| None | 31 (41.3) | 173 (47.8) |  |  |
| 1–2 times | 25 (33.3) | 83 (22.9) |  |  |
| 3–4 times | 14 (18.7) | 24 (6.6) |  |  |
| 5 or more times | 13 (17.3) | 88 (24.3) |  |  |
| **Number of long-term medication types** |  |  | 50.351 | <0.001 |
| No long-term medication | 12 (16.0) | 206 (56.9) |  |  |
| 5 or fewer medications | 46 (61.3) | 138 (38.1) |  |  |
| More than 5 medications | 17 (22.7) | 18 (5.0) |  |  |
| **Frequency of long-term medication use** |  |  | 55.147 | <0.001 |
| No long-term medication | 14 (18.7) | 203 (56.1) |  |  |
| Once a day | 15 (20.0) | 61 (16.9) |  |  |
| Twice a day | 12 (16.0) | 53 (14.6) |  |  |
| Three times or more a day | 34 (45.3) | 45 (12.4) |  |  |
| **Duration of long-term medication use** |  |  | 45.084 | <0.001 |
| No | 7 (9.3) | 187 (51.7) |  |  |
| Yes | 68 (90.7) | 175 (48.3) |  |  |
| **Number of comorbidities** |  |  | 57.278 | <0.001 |
| 1≤ | 25 (31.3) | 278 (26.1) |  |  |
| 2 | 26 (37.3) | 52 (14.7) |  |  |
| ≥3 | 24 (28.9) | 32 (9.2) |  |  |
| **Social frailty** |  |  | 46.642 | <0.001 |
| Yes | 34 (30.1) | 44 (12.5) |  |  |
| No | 41 (69.9) | 318 (87.5) |  |  |
| **EQ-5D-5L trade-off value** | 0.86 (0.56, 0.94) | 1.0 (0.89, 1) | −7.409 | <0.001 |

**Note:**
Asterisk (*) indicates the use of Fisher's exact test.

multimorbidity, social frailty, and poor QOL as significant factors associated with PPC multimorbidity.

The prevalence of PPC multimorbidity in our cohort was 17.2%, closely matching the 18% reported by *He et al. (2024)* in a Chinese middle-aged and older population. We sampled only community-dwelling adults aged ≥ 60 years, applied a broader list of chronic diseases, and used face-to-face cognitive assessments, in contrast to their inclusion of adults aged 45+ with seven predefined conditions and telephone-based screening. By comparison, our prevalence is well above the 6% pooled estimate from a multinational meta-analysis by *Ni et al. (2023)* (seven studies, 2017–2020)—a difference that probably
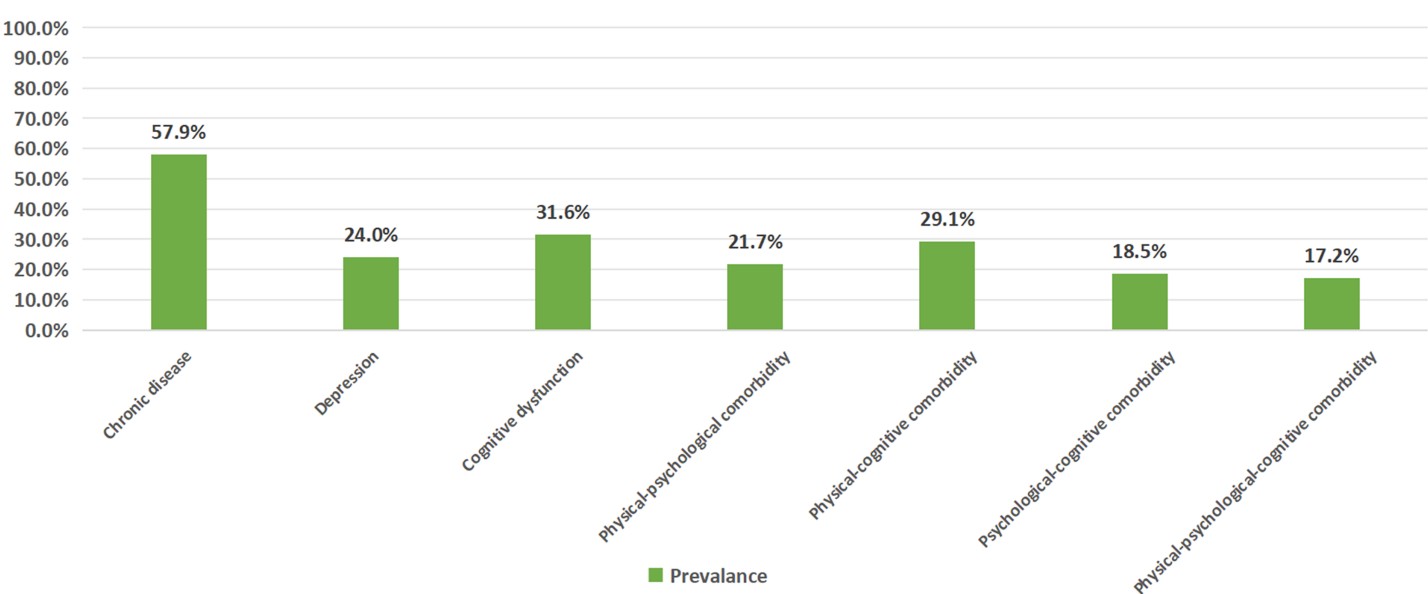

**Figure 1 Prevalence of physical, psychological, and cognitive comorbidities.**

**Table 2 Results of multivariate logistic regression analysis ($n$ = 437).**

| Variables | $P$ value | Odds radio | 95% Confidence interval | |
|---|---|---|---|---|
| | | | Lower bound | Upper bound |
| Long-term medication use (No) | Reference | | | |
| Long-term medication use (Yes) | 0.013 | 3.240 | 1.279 | 8.208 |
| Number of comorbidities (≤1) | Reference | – | – | – |
| Number of comorbidities (=2) | 0.001 | 3.635 | 1.699 | 7.774 |
| Number of comorbidities (≥3) | <0.001 | 7.312 | 3.267 | 16.362 |
| Social frailty (No) | Reference | | | |
| Social frailty (Yes) | <0.001 | 3.493 | 1.739 | 7.014 |
| EQ-5D score | <0.001 | 0.072 | 0.020 | 0.257 |
| Constant | 0.072 | 0.277 | – | – |

stems from the older mean age (72.0 years) and socio-economic and cultural factors specific to contemporary China. The high prevalence of PPC multimorbidity in this population underscores the urgent need for healthcare and social service professionals to pay greater attention to this issue. Given the significant negative impact of PPC comorbidity on quality of life, functional decline, and increased healthcare burden, it is essential for both healthcare providers and policymakers to implement targeted interventions and care strategies that address the unique needs of older adults with PPC multimorbidity.

Our findings highlight social frailty as a significant contributor to PPC multimorbidity in community-dwelling older adults in China, supporting global evidence on the role of social disconnection in accelerating multisystem health decline (*Foster et al., 2023*;

*Luo, 2023*). The erosion of traditional intergenerational support systems—evidenced by high rates of solitary living—is likely associated with biological stress responses, which in turn impair cognitive and psychological resilience (*Guarnera, Yuen & Macpherson, 2023*). Social frailty also impedes adherence to health-promoting behaviours and delays care-seeking for emerging health concerns, emphasizing the need for community-based nursing strategies that prioritize social connectivity as a key metric in geriatric assessments (*Li et al., 2023*). Culturally sensitive interventions, such as intergenerational co-living initiatives and rural telehealth networks, could help mitigate health deterioration driven by social isolation while respecting cultural norms like filial piety (*Kim et al., 2024*; *Zhang, Leuk & Teo, 2023*). These strategies could address the critical gap in support for older adults living alone, enhancing social resilience and reducing the negative effects of social frailty.

The lower mean EQ-5D-5L trade-off values in the physical-psychological-cognitive multimorbidity group align with the established link between PPC multimorbidity and reduced quality of life (*Tran et al., 2022*; *Liang et al., 2024*). This emphasizes the need for a comprehensive approach to care that addresses both physical and psychosocial well-being. The interplay between physical, psychological, and cognitive frailties increases the complexity of managing these elderly and contributes to functional decline and greater care dependency (*Sieber et al., 2023*). Interventions targeting physical rehabilitation, mental health support, and cognitive function, alongside routine assessments of these factors, could improve outcomes (*Yang et al., 2022*). Additionally, community-based models that promote social engagement and cognitive resilience may help mitigate the compounded effects of these comorbidities, ultimately enhancing quality of life (*Crocker et al., 2024*).

The challenges of polypharmacy in elderly care are underscored by our findings regarding long-term medication use. The use of multiple medications and extended treatment regimens increases the risk of adverse drug events, compromises medication adherence, and impacts overall health (*Hargraves & Montori, 2019*; *Zhu et al., 2023*). These issues stress the importance of careful medication management, proactive monitoring for adverse effects, and deprescribing strategies where clinically appropriate to optimize outcomes in this vulnerable population (*Yang et al., 2022*). The increasing number of comorbidities is significantly correlated with the development of PPC multimorbidity, a finding that is consistent with the results of previous studies (*Zhao et al., 2024*; *Posis et al., 2024*). As the number of comorbidities increases, so does the complexity of managing elderly patients, particularly those with multiple chronic conditions (*Li et al., 2024*). This underscores the need for a comprehensive, patient-centered approach that integrates physical, psychological, and cognitive care, addressing the interconnected needs of these individuals to improve overall health outcomes (*Kivelitz et al., 2021*).

This study benefits from the use of validated instruments to comprehensively assess various health aspects, employing multivariate analysis to identify independent predictors of PPC multimorbidity. However, certain limitations should be acknowledged. This study has several limitations. First, its cross-sectional design precludes causal inferences between associated factors and PPC multimorbidity. Second, the use of convenience sampling may

have led to selection bias, limiting the generalizability of findings. Third, the relatively small and homogeneous sample further restricts applicability to broader populations and healthcare settings. Additionally, reliance on self-reported measures may introduce recall and social desirability biases. Future research should employ longitudinal designs, probability sampling, and incorporate objective assessments to strengthen the validity and generalizability of findings. These insights may inform the development of targeted interventions for older adults with multimorbid conditions.

## CONCLUSIONS

This study underscores the burden of PPC multimorbidity in community-dwelling older adults in China, highlighting key factors such as long-term medication use, advanced multimorbidity, social frailty, and poor quality of life. These findings call for a comprehensive, patient-centered care approach, with an emphasis on addressing social frailty and polypharmacy. Future research should explore these relationships further through longitudinal studies with larger, more diverse samples to inform targeted interventions for improved health outcomes in the elderly.

## ACKNOWLEDGEMENTS

During the preparation of this work the authors used ChatGPT in order to improve the readability and language of the manuscript. After using this tool, the authors reviewed and edited the content as needed and take full responsibility for the content of the published article.

### Funding
The authors received no funding for this work.

### Competing Interests
The authors declare that they have no competing interests.

### Author Contributions
- Lin Lin conceived and designed the experiments, performed the experiments, analyzed the data, prepared figures and/or tables, authored or reviewed drafts of the article, and approved the final draft.
- Di-fei Duan conceived and designed the experiments, performed the experiments, analyzed the data, prepared figures and/or tables, authored or reviewed drafts of the article, and approved the final draft.
- Linjia Yan conceived and designed the experiments, performed the experiments, analyzed the data, prepared figures and/or tables, authored or reviewed drafts of the article, and approved the final draft.
- Hai yan He conceived and designed the experiments, authored or reviewed drafts of the article, and approved the final draft.

## Human Ethics

The following information was supplied relating to ethical approvals (*i.e.*, approving body and any reference numbers):

Ethical Review Committee of Biomedical Research, West China Hospital, Sichuan University (Approval No. 2024(2475)).

## Data Availability

The raw data are available in the Supplemental File.

## Supplemental Information

Supplemental information for this article can be found online at http://dx.doi.org/10.7717/peerj.19750#supplemental-information.

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
