# Peer review of "Prevalence and associated factors of physical-psychological-cognitive multimorbidity in Chinese community-dwelling older adults: a cross-sectional study"

_PeerJ, doi:10.7717/peerj.19750_

## Round 0.1 · original submission · Minor Revisions

Thank you for submitting your manuscript entitled "Prevalence and Associated Factors of Physical-Psychological-Cognitive Multimorbidity in Older Adults in the Chinese Community: A Cross-Sectional Study" to our journal. After a thorough evaluation by two expert reviewers, we are pleased to inform you that the manuscript has been deemed a relevant and timely contribution to the study of physical-psychological-cognitive (PPC) multimorbidity in older adults. The study is well-designed, employs validated instruments, and presents findings with significant public health implications.

Based on the reviewers' reports, the editorial decision is to accept the manuscript pending minor revisions.

Both reviewers highlight the overall strength of the study, particularly the comprehensive approach that simultaneously addresses physical, psychological, and cognitive domains, as well as the identification of key predictors such as social frailty, prolonged medication use, and quality of life. The use of validated tools and the practical implications for interventions are especially valued.

However, the reviewers also noted several methodological and editorial aspects that should be addressed to improve the clarity, transparency, and overall quality of the manuscript. These issues do not substantially affect the validity of the study but should be revised prior to publication.
Key comments to be addressed include:

Rephrase expressions in the Introduction and Discussion sections that may imply causality, particularly those related to social frailty.

Please emphasize that the findings are correlational due to the cross-sectional design.

Improve the writing to avoid unnecessary repetition and clarify the main research question.

Clearly define the operational criteria used to classify PPC multimorbidity, including the thresholds applied and whether the coexistence of all three dimensions was required.

Justify the cut-off points used for the PHQ-9 and the HALFT scale, especially with regard to their internal consistency.

Include information on how missing or inconsistent data were handled.
Add a clear data availability statement, specifying that data will be publicly accessible through the journal’s repository upon publication.

Incorporate confidence intervals for odds ratios (ORs) and effect sizes in the statistical analyses.

Correct minor typographical errors, especially in the references (spacing, punctuation, duplicated brackets).

Ensure consistency in the reference formatting throughout the manuscript.
We look forward to receiving your revised manuscript addressing these points.

Please include a point-by-point response to reviewers with your submission.
Kind regards

Reviewer 1 ·

Basic reporting

The manuscript is generally well written and professionally structured. Nevertheless, we recommend reviewing certain stylistic elements to improve clarity and conciseness. For instance, in the Introduction:

“The increasing prevalence of multimorbidity, particularly physical-psychological-cognitive (PPC) multimorbidity, poses significant challenges to older adults.”
This sentence could be revised to avoid the immediate repetition of “multimorbidity,” which may affect readability.

In addition, we observed minor issues in the formatting and punctuation of reference citations. These include inconsistencies in spacing before brackets and errors in bracket placement or duplication, such as in reference lines on pages 8, 15, and 16. A careful review of these aspects throughout the manuscript is advisable to ensure consistency and typographical accuracy.

Overall, the manuscript meets PeerJ’s basic reporting standards, with a few minor issues that can be easily addressed.

Experimental design

1. Research Question and Operational Definitions
While the research aim is clear, the Introduction would benefit from explicitly stating the main research question. For example, a direct formulation such as “What is the prevalence of PPC multimorbidity and its associated factors among community-dwelling older adults in China?” could improve focus and clarity.

In addition, the operational definition of PPC multimorbidity is not clearly articulated. It would be helpful to specify the exact criteria used to classify a participant as having PPC—namely, what thresholds were applied to define physical, psychological, and cognitive conditions, and whether all three were required concurrently.

2. Methodological Transparency
The use of validated tools such as the PHQ-9, AD-8, EQ-5D-5L, and HALFT is appropriate and appreciated. However, some additional explanation would enhance clarity. For example, while the manuscript uses a PHQ-9 cut-off of ≥5 to indicate depressive symptoms, it would be useful to include a reference or brief justification supporting that threshold.

The HALFT scale, although practical for community-based assessments of social frailty, is reported with a Cronbach’s alpha of 0.602. This falls within the lower range of internal consistency. A brief rationale for its use—perhaps referencing prior validation studies or its feasibility in low-resource settings—would help address this limitation.

The sample size calculation is now clearly described in the updated manuscript, including assumptions and statistical parameters. This is appreciated and meets reporting standards.

However, one important methodological detail that is missing is how missing or inconsistent data were handled. Clarifying whether any imputation or exclusion criteria were applied would strengthen the transparency of the analysis.

3. Sampling and Generalizability
The use of convenience sampling is understandable in community settings, but it inherently limits the generalizability of findings. It would be helpful for the authors to briefly acknowledge this limitation in the Discussion, particularly in relation to selection bias (e.g., differences between those who participated and those who declined).

4. Reproducibility and Supplementary Materials
The supplementary materials—particularly the recruitment flowchart (Supplementary Figure 1) and the EQ-5D-5L utility value table (Supplementary Table 1)—are appropriate and relevant for interpreting the methodology. These documents are referenced in the manuscript, and it is assumed that they will be made accessible upon publication. To enhance clarity, the authors might consider more explicitly describing how these materials contribute to the study (e.g., referencing the recruitment flowchart when discussing participant selection, or briefly explaining how utility scores were derived from Supplementary Table 1).

5. Formatting and Consistency
There are still some inconsistencies in how references are formatted, particularly in the spacing around citation brackets. Additionally, a typographical error involving double brackets (e.g., “[[35, 36]”) appears in the Discussion. A careful review of reference formatting throughout the manuscript is recommended.

These clarifications will help improve the transparency and reproducibility of the study.

Validity of the findings

The manuscript presents a well-designed and statistically sound cross-sectional analysis of PPC multimorbidity among older adults in China. The findings are relevant and derived from validated tools and appropriate statistical methods. However, a few clarifications would improve transparency and alignment with PeerJ’s standards.

1. Replication and Contribution to the Literature
The study does not rely on claims of novelty or impact, aligning with PeerJ’s scope. While it is not a direct replication, it extends prior research by examining PPC multimorbidity in a Chinese population using validated methods. To strengthen this contribution, the authors could briefly elaborate on how their findings complement or contrast with previous work (e.g., He et al., 2024), particularly regarding cultural or socioeconomic factors unique to China.

2. Data Quality and Accessibility
The analyses are robust and statistically controlled, with validated instruments (PHQ-9, AD-8) ensuring reliability. Multivariable logistic regression was applied appropriately to identify predictors.

Data Accessibility: The manuscript references supplementary materials (Figure S1, Table S1), which support replicability. Since the underlying data have been submitted to the journal, it is expected that they will be made available upon publication. For transparency, the authors should include a data availability statement in the manuscript (e.g., “All raw data will be publicly accessible through PeerJ’s repository upon publication”).

3. Strength of the Conclusions
The conclusions are generally well-supported and avoid causal claims, appropriately acknowledging the cross-sectional design. However:

Language Precision: A few phrases in the Discussion (e.g., “social frailty exacerbates biological stress responses”) could imply causation. Recommend revising to emphasize correlation (e.g., “social frailty is associated with biological stress responses”).

Statistical Reporting: Including confidence intervals for all odds ratios (e.g., OR = 3.24, 95% CI: 1.28–8.21) and effect sizes (e.g., Nagelkerke’s R²) would contextualize the clinical relevance of predictors.

Conclusion
The findings are valid, well-supported, and clearly presented. With minor clarifications—such as a data availability statement, refined causal language, and enhanced statistical reporting—the manuscript will fully meet PeerJ’s standards for transparency and interpretative rigor.

Additional comments

This manuscript addresses an important and timely topic in geriatric public health, offering valuable insight into the prevalence and correlates of physical-psychological-cognitive (PPC) multimorbidity among older adults in China. The study is well-conceived and presented with a clear structure, appropriate methodology, and thoughtful discussion.

The recommendations provided throughout the review are intended to help the authors further clarify methodological details, refine interpretative language, and enhance overall transparency. With these minor revisions, the manuscript will be well-positioned to contribute meaningfully to the field and meet the expectations for publication in PeerJ.

·

Basic reporting

The study examining the prevalence and predictors of physical-psychological-cognitive (PPC) multimorbidity showcases several strengths along with some notable weaknesses. One of its main strengths is the comprehensive definition of PPC multimorbidity, which highlights the interconnectedness of physical, psychological, and cognitive health issues. The reported prevalence of 17.2% for PPC multimorbidity, alongside distinct rates for other combinations of conditions, emphasizes the complexity of health challenges individuals face. Additionally, the identification of significant predictors such as long-term medication use, social frailty, and lower quality of life (measured through EQ-5D scores) offers valuable insights for targeted healthcare interventions, reinforcing the need for integrated approaches that consider a patient’s overall health landscape.

The study also effectively addresses factors like social frailty and living conditions, aligning with current trends in healthcare that prioritize patient-centered care. Its recommendations for innovative interventions, like promoting intergenerational living arrangements and utilizing telehealth, demonstrate a proactive approach to addressing these challenges in real-world settings.

However, there are some weaknesses to consider. The sample size of 437 participants may limit the study's generalizability, as larger, more diverse samples could provide stronger evidence. Additionally, the cross-sectional design of the study restricts the ability to confirm causal relationships—while it identifies associations between predictors and PPC multimorbidity, it does not clarify whether these factors lead to multimorbidity or are consequences of it. Longitudinal studies would therefore be beneficial in establishing clearer cause-and-effect dynamics.

Another limitation is the reliance on self-reported measures, such as the EQ-5D and assessments of social frailty, which can introduce biases; individuals may misreport their health or social situation. Using more objective assessments could provide a more accurate picture of the participants' health.

In summary, while the study contributes significantly to understanding PPC multimorbidity through detailed analysis and actionable recommendations, it also has limitations related to sample size, study design, and data collection methods. Addressing these weaknesses in future research would enhance our understanding and lead to better healthcare strategies for those experiencing multimorbid conditions.

Experimental design

The study on physical-psychological-cognitive (PPC) multimorbidity presents several strengths and weaknesses. On the positive side, it evaluates multiple health dimensions—physical, psychological, and cognitive—offering a comprehensive view of multimorbidity. Additionally, the study effectively explores various predictors that enhance the understanding of what influences PPC multimorbidity, employing reliable measures such as the EQ-5D for assessing quality of life. However, the study also has notable limitations. Its cross-sectional design captures data at a single point, which restricts the ability to draw causal inferences about the relationships between predictors and PPC multimorbidity. Furthermore, the sample size of 437 participants may not be large or diverse enough to allow for generalizable findings. The reliance on self-reported data introduces potential biases that can affect the accuracy of the results. Lastly, the absence of longitudinal data prevents insights into how PPC multimorbidity evolves. In summary, while the study contributes valuable insights into PPC multimorbidity, its design limitations underscore the need for larger and longitudinal research in this field to deepen understanding.

Validity of the findings

The study on physical-psychological-cognitive (PPC) multimorbidity has some strengths but also important weaknesses. While it uses reliable tools for measuring health, the way the study is set up makes it hard to say what causes what because it looks at data at just one time instead of over a longer period. Also, since it relies on people's self-reports, there could be inaccuracies. With only 437 participants, the findings may not apply well to larger or more diverse groups, limiting their usefulness. Overall, the study provides helpful insights, but future research should include more participants and look at changes over time to better understand PPC multimorbidity.

---

## Round 0.2 · accepted · Accept

Dear Dr. He,
Your manuscript has been accepted for publication. Congratulations!

Reviewer 1 ·

Basic reporting

The authors have adequately addressed all the requested revisions, and the manuscript has been substantially improved as a result.

Experimental design

The authors have adequately addressed all the requested revisions, and the manuscript has been substantially improved as a result.

Validity of the findings

The authors have adequately addressed all the requested revisions, and the manuscript has been substantially improved as a result.

Additional comments

The authors have adequately addressed all the requested revisions, and the manuscript has been substantially improved as a result.